# Dual-Specificity Protein Phosphatase 4 (DUSP4) Overexpression Improves Learning Behavior Selectively in Female 5xFAD Mice, and Reduces β-Amyloid Load in Males and Females

**DOI:** 10.3390/cells11233880

**Published:** 2022-12-01

**Authors:** Allen L. Pan, Mickael Audrain, Emmy Sakakibara, Rajeev Joshi, Xiaodong Zhu, Qian Wang, Minghui Wang, Noam D. Beckmann, Eric E. Schadt, Sam Gandy, Bin Zhang, Michelle E. Ehrlich, Stephen R. Salton

**Affiliations:** 1Nash Family Department of Neuroscience, Icahn School of Medicine at Mount Sinai, One Gustave L. Levy Place, New York, NY 10029, USA; 2Department of Neurology, Icahn School of Medicine at Mount Sinai, One Gustave L. Levy Place, New York, NY 10029, USA; 3Department of Psychiatry, Icahn School of Medicine at Mount Sinai, One Gustave L. Levy Place, New York, NY 10029, USA; 4Department of Genetics and Genomic Sciences, Icahn School of Medicine at Mount Sinai, One Gustave L. Levy Place, New York, NY 10029, USA; 5Mount Sinai Center for Transformative Disease Modeling, Icahn School of Medicine at Mount Sinai, One Gustave L. Levy Place, New York, NY 10029, USA; 6Department of Psychiatry and Alzheimer’s Disease Research Center, Icahn School of Medicine at Mount Sinai, New York, NY 10029, USA; 7Department of Pediatrics, Icahn School of Medicine at Mount Sinai, New York, NY 10029, USA; 8Brookdale Department of Geriatrics and Palliative Medicine, Icahn School of Medicine at Mount Sinai, New York, NY 10029, USA

**Keywords:** Alzheimer’s disease, dual-specificity protein phosphatase 4, disease-associated microglia, mitogen-activated protein kinase, neuroinflammation

## Abstract

Recent multiscale network analyses of banked brains from subjects who died of late-onset sporadic Alzheimer’s disease converged on *VGF* (non-acronymic) as a key hub or driver. Within this computational *VGF* network, we identified the dual-specificity protein phosphatase 4 (*DUSP4*) [also known as mitogen-activated protein kinase (MAPK) phosphatase 2] as an important node. Importantly, *DUSP4* gene expression, like that of *VGF*, is downregulated in postmortem Alzheimer’s disease (AD) brains. We investigated the roles that this *VGF*/*DUSP4* network plays in the development of learning behavior impairment and neuropathology in the 5xFAD amyloidopathy mouse model. We found reductions in *DUSP4* expression in the hippocampi of male AD subjects, correlating with increased CDR scores, and in 4-month-old female and 12–18-month-old male 5xFAD hippocampi. Adeno-associated virus (AAV5)-mediated overexpression of DUSP4 in 5xFAD mouse dorsal hippocampi (dHc) rescued impaired Barnes maze performance in females but not in males, while amyloid loads were reduced in both females and males. Bulk RNA sequencing of the dHc from 5-month-old mice overexpressing DUSP4, and Ingenuity Pathway and Enrichr analyses of differentially expressed genes (DEGs), revealed that DUSP4 reduced gene expression in female 5xFAD mice in neuroinflammatory, interferon-gamma (IFNγ), programmed cell death protein-ligand 1/programmed cell death protein 1 (PD-L1/PD-1), and extracellular signal-regulated kinase (ERK)/MAPK pathways, via which DUSP4 may modulate AD phenotype with gender-specificity.

## 1. Introduction

DUSP4 is a member of the dual-specificity protein phosphatase (DUSP) family that regulates mitogen-activated protein kinase (MAPK) activity [1]. MAPK signaling pathways regulate important biological processes including cell proliferation, cell death, and cell survival [2,3,4]. DUSP4 contains a consensus protein kinase-interacting motif (KIM) that allows it to interact with the docking domain of MAPKs [5], and a catalytic site that consists of conserved Asp, Arg, and Cys residues [6]. DUSP4 is localized in the nucleoplasm [7] and has been reported to selectively dephosphorylate extracellular signal-regulated protein kinase(s) (ERKs) and c-Jun N-terminal kinase(s) (JNKs) in vitro [8]. The roles of DUSP4 in neuronal function have not been extensively studied, but accumulating evidence suggests that DUSPs are linked to psychiatric and neurological disorders [9,10]. Indeed, the level of DUSP4 was reported to be downregulated in the postmortem brains of Alzheimer’s disease (AD) subjects [11], and previous network analyses identified DUSP4 as a hub gene in a major depressive disorder (MDD) female-specific module, which resulted in ERK activation [12]. It remains to be fully investigated whether decreases in DUSP4 and/or VGF levels in MDD and AD [13,14] could underlie shared pathophysiological mechanisms that drive the comorbidity that has frequently been observed with these two highly burdensome diseases, but is not explained by common genetic variants [15]. The association of DUSP4 with AD was further implicated in studies from animal models of disease-related pathologies, showing that germline *Dusp4* gene ablation in knockout mice led to spatial reference and working learning behavior deficits [16].

One of the pathological hallmarks of AD is the accumulation and deposition of amyloid-β peptide (Aβ). Aβ is generated by the sequential proteolytic processing of the transmembrane protein amyloid precursor protein (APP) via β- and γ-secretases. Multiple studies have shown that MAPK signaling pathways are involved in AD pathology through their regulation of β-secretase (beta-APP-site-cleavage enzyme-1; BACE1) and γ-secretase activities [17,18]. DUSP4 has been shown to regulate neuronal differentiation through ERK dephosphorylation [19], but there are no studies to date that implicate DUSP4 in Aβ-induced cytotoxicity.

Through a computational network-based approach, we previously identified DUSP4, within the VGF network, as a potential mediator of AD pathogenesis [14]. DUSPs are regulated by growth factors including brain-derived neurotrophic factor (BDNF) and nerve growth factor (NGF), both of which promote DUSP transcription and post-translational stabilization of the DUSP protein, and regulate neuronal survival and differentiation [20]. NGF induces DUSP4 gene expression in vitro [21,22], while VGF gene and protein expression are induced by several neurotrophic factors including BDNF [23] and NGF [24]. The VGF proprotein is processed into multiple neuropeptides that have different physiological functions including the regulation of synaptic plasticity, which is BDNF-dependent in the hippocampus [25]. VGF-derived neuropeptides have been shown to regulate learning behavior, neuroprotection, and neuroinflammation in AD [26,27]. Interestingly, VGF is one of the most downregulated proteins found in brain tissues from AD patients [28], highlighting the potential importance of VGF and the VGF network in AD.

DUSP4 is located downstream of VGF in the VGF network [14], and to better understand how DUSP4 functions in the multiscale network and contributes to AD pathogenesis and progression, we overexpressed DUSP4 in the dorsal hippocampus (dHc) of the 5xFAD mouse model of Alzheimer’s amyloidosis. Administration of AAV5-DUSP4 to 5xFAD dHc led to sex-dependent improvements in Barnes maze spatial learning behavior that were associated with reductions in dHc amyloid burden. DUSP4 overexpression did not lead to widespread changes in APP secretase-processing enzyme mRNA levels in 5xFAD dHc, nor did it increase expression of VGF network hub genes, including *Vgf* itself, *Sst*, *Bdnf* and *Scg2*, all of which are reduced in AD [14]. However, importantly, RNA sequencing (RNAseq) analysis identified neuroinflammatory signaling pathways that were downregulated by DUSP4 overexpression in female 5xFAD mice, suggesting that DUSP4 regulation of these pathways might be involved in AD-associated neuropathology. These pathways included interferon-gamma (IFNγ), ERK/MAPK, and programmed cell death protein 1/programmed cell death protein-ligand 1 (PD-L1/PD-1).

## 2. Materials and Methods

### 2.1. Animals

All experimental procedures were conducted in accordance with the NIH guidelines for animal research and were approved by the Institutional Animal Care and Use Committee (IACUC) at the Icahn School of Medicine at Mount Sinai (ISMMS). All mice were housed on a 12 h light-dark cycle with ad libitum access to food and water. 5xFAD transgenic mice that overexpress human APP695 with Swedish (K670N, M671L), Florida (I716V) and London (V717I) familial AD (FAD) mutations and human presenilin1 (PSEN1) with the M146L and L286V FAD mutations [29] were obtained from Jackson Labs (Bar Harbor, ME; JAX#34840) and were maintained on a mixed B6/SJL genetic background as described [14]. Female and male wild-type (WT) and 5xFAD mice at 4 months of age were anesthetized by intraperitoneal injection of ketamine (100 mg/kg) and xylazine (10 mg/kg) mixture per 10 g of body weight, and were infused bilaterally using a twenty-five gauge needle (Hamilton, Reno, NV) with 1.0 μL of AAV5-GFP, or AAV5-DUSP4 (4 × 10^12^ vg/mL) into dHc (AP = −2.0 mm, ML = ± 1.5 mm, and DV = −2.0 mm relative to the bregma) at a rate of 0.2 μL per min. The needle was allowed to remain in place for three minutes before removal to prevent backflow. AAV5-injected mice were allowed to recover for a month before behavioral testing. AAV5-GFP (control) and AAV5-mouse DUSP4 (VectorBuilder Inc., Chicago, IL, USA; AAV-5′ITR-CAG-mDUSP4-WPRE-BGHpA-3′ITR) (AAV5 serotype/AAV2 genotype) were prepared by the Vector Core at the University of North Carolina at Chapel Hill.

### 2.2. Barnes Maze Test

A standard apparatus was used for the Barnes Maze test [30], as described [31]. Briefly, 5-month-old mice were transferred to a closed starting chamber at the center of the platform for 10 s prior to exploring the maze for 3 min. The experimenter would guide any mice that failed to enter the escape box within 3 min, and the latency was recorded as 180 s. Then the mice were allowed to remain in the escape box for 1 min before the next trial. The experiments consisted of two trials per day for 5 consecutive days. The platform and the escape box were cleaned with 70% ethanol to eliminate the potential olfactory cues after each trial. All trials were recorded by video camera and analyzed with ANY-maze video tracking software (Stoelting Co., Wood Dale, IL, USA).

### 2.3. Tissue Collection and Sample Preparation

Mice were anesthetized in a CO_2_ chamber and then transcardially perfused with 20 mL ice-cold phosphate buffered saline (PBS). The right hemisphere was post-fixed by incubation for 24 h in 4% PFA followed by impregnation in 30% sucrose until the brains sunk to the bottom. Fixed brains were then cut into 30 µm coronal sections with a cryostat (Leica, Deer Park, IL, USA). The contralateral hemisphere was dissected to isolate dHc, which was hemi-sected: half of the dHc was homogenized in radioimmunoprecipitation assay buffer (RIPA; Sigma-Aldrich, Burlington, MA, USA, R0278; 50 mM Tris-HCl, pH 8.0, 150 mM sodium chloride, 1.0% NP-40, 0.5% sodium deoxycholate, 0.1% sodium dodecyl sulfate) containing phosphatase and protease inhibitors (Roche, South San Francisco, CA, USA), was centrifuged for 20 min at 15,000× *g*, and the supernatant collected. The other half of the brain was used for RNA extraction using the RNeasy Mini Kit (Qiagen, Germantown, MD, USA). A separate cohort of mice was anesthetized and perfused with PBS as described above, and the entire hippocampus was isolated and lysed in 8M urea, 10 mM Tris, 100 mM NaH_2_PO4, pH 8.5, supplemented with HALT protease and phosphatase inhibitor cocktail (Thermo Fisher Scientific, Asheville, NC, USA), as described in [32].

### 2.4. RNA Extraction and Quantitative Real-Time PCR Analysis

RNAs were isolated from hippocampi using the QIAzol^®^ Lysis Reagent (Qiagen, Germantown, MD, USA) and the miRNeasy^®^ Mini Kit (Qiagen, Germantown, MD, USA) following the manufacturer’s instructions. The concentration and the purities of the total RNA samples were spectrophotometrically assessed at 260 and 280 nm using NanoDrop 2000c (Thermo Fisher Scientific, Asheville, NC, USA). Then the total RNA was reverse transcribed into cDNA in a 20 µL reaction volume using a high-capacity RNA-to-cDNA^TM^ kit (Applied Biosystems, Dallas, TX, USA; 4387406). Real-time PCR reaction was performed on a QuantStudio 7 Flex Real-Time PCR System (Applied Biosystems, Dallas, TX, USA) with the Maxima SYBR Green/ROX qPCR Master Mix (Thermo Fisher Scientific, Atlanta, GA, USA; K0221). Reactions were carried out in duplicate in 384-well plates (Applied Biosystems, Dallas, TX, USA) according to the manufacturer’s three-step cycling protocol. To perform RT-PCR analyses, the relative gene expression of each transcript was normalized to the reference gene *Gapdh* with ΔCt method. The sequences of oligonucleotides used:

*Gapdh* 5′-AACGACCCCTTCATTGACCT-3′ and

5′-TGGAAGATGGTGATGGGCTT-3′,

Mouse *Dusp4* 5′-CCTGCTTAAAGGTGGCTATGAGA-3′ and

5′-GGTGCTGGGAGGTACAGGG-3′,

Human *DUSP4* 5′-GGCATCACGGCTCTGTTGAAT-3′ and

5′-GTCGGCCTTGTGGTTATCTTC-3′,

*Vgf* 5′-CGAAGAAGCAGCAGAAGCTC-3′ and

5′-TCGAAGTTCTTGGAGCAAGG-3′,

*Sst* 5′-CCGTCAGTTTCTGCAGAAGT-3′ and

5′-CAGGGTCAAGTTGAGCATCG-3′,

*Bdnf* 5′-GAAGAGCTGCTGGATGAGGAC-3′ and

5′-CGAGTTCCAGTGCCTTTTGTC-3′,

*Scg2* 5′- AGGGTTGACGAGGAACAAA-3′ and

5′-CTGGACTGGGCACTCTCTTC-3′,

Mouse *Psen1* 5′-CAAAAACAGAGAGCAAGCCC-3′ and

5′-TCTCTCAAGTCACTGAGGGACA-3′,

Human *PSEN1* 5′-GCAGTATCCTCGCTGGTGAAGA-3′ and

5′-CAGGCTATGGTTGTGTTCCAGTC-3′,

*Ps2* 5′-CTGGTGTTCATCAAGTACCTGCC-3′ and

5′-TTCTCTCCTGGGCAGTTTCCAC-3′,

*Adam10* 5′-TAAGGAATTATGCCATGTTTGCTGC-3′ and

5′-ACTGAACTGCTTGCTCCACTGCA-3′,

*Adam17* 5′-TTGGAGCAGAACATGACCCTGATGG-3′ and

5′-TGCAGCAGGTGTCGTTGTTCAGGTA-3′,

*Bace1* 5′-TCTTTTCCCTGCAGCTCTGT-3′ and

5′-ACTGCCCGTGTATAGCGAGT-3′

*Nct* 5′-CCAAGCTTCCCATTGTGTGC-3′ and

5′-TGCTGAAGGTGCTCTGGATG-3′

*Aph1a* 5′-GTGCTGCTGTCTCTGTCCTT-3′ and

5′-TCTGTCGGATGGAGATGGGT-3′

*Aph1b* 5′-CTGGGGCGTTGTGTTCTTTG-3′ and

5′-AAATGCCCAGATGCCCATGA-3′

*Aph1c* 5′-TTCCTCATCGCTGGTGCTTT-3′ and

5′-CGCTCCGAAGATGAGCAGAT-3′

*Stat1* 5′-GCCTCTCATTGTCACCGAAGAAC-3′ and

5′-TGGCTGACGTTGGAGATCACCA-3′

*Elf* 5′-ATGCTTGCCAGCCCACTACAGA-3′ and

5′-CCATTGGTCAGCACCGTAGTCA-3′

*Rac2* 5′-CTCAGCCAATGTGATGGTGGAC-3′ and

5′-CGGACATTCTCATAGGAGGCTG-3′

*Dusp2* 5′-AGATGGTGGAGATAAGTGCCTGG-3′ and

5′-AGATGGTGGCTGAGCGAGAGAT-3′

*Rps6ka1* 5′-TATGGAGCCTGGGCATTCTGCT-3′ and

5′-GCTGTCTCTGAAACCGTGTTCC-3′

*B2m* 5′-ACAGTTCCACCCGCCTCACATT-3′ and

5′-TAGAAAGACCAGTCCTTGCTGAAG-3′

*Ccl5* 5′-CCTGCTGCTTTGCCTACCTCTC-3′ and

5′-ACACACTTGGCGGTTCCTTCGA-3′

*Ccl12* 5′-GCTACAGGAGAATCACAAGCAGC-3′ and

5′-ACGTCTTATCCAAGTGGTTTATGG-3′

*Icam1* 5′-AAACCAGACCCTGGAACTGCAC-3′ and

5′-GCCTGGCATTTCAGAGTCTGCT-3′

*Cd68* 5′-TCCAAGCCCAAATTCAAATC-3′ and

5′-ATGGGTACCGTCACAACCTC-3′

*Iba1* 5′-GTCGCACTCAGCAACAGG-3′ and

5′-ACTTCTGGTCACAGAGGAACTC-3′

### 2.5. Immunohistochemistry

Coronal sections obtained from the right hemisphere (30 μm thickness) were used for immunostaining. For each mouse brain, three sections of tissue covering the dHc were randomly selected. The coronal sections were first washed with PBS, and then incubated with the following primary antibodies in 0.1% Triton X-100 in PBS overnight at 4 °C: anti-DUSP4 (1:500, Abcam, Cambridge, MA, USA; ab216576); anti-Ab (1:500, Biolegend, San Diego, CA, USA; 803001); anti-GFAP (1:1000, Abcam, Cambridge, MA, USA; ab53554); or anti-NeuN (1:1000, Invitrogen, Atlanta, GA, USA; MA5-33103). Sections were rinsed with PBS and incubated for 1 h at room temperature with appropriate secondary antibodies: anti-rabbit Alexa Fluor IgG 488 or 568 (1:1000, Invitrogen, Atlanta, GA, USA) and anti-mouse IgG Alexa 488 (1:1000, Invitrogen, Atlanta, GA, USA). Sections were then washed with PBS, allowed to air-dry in the dark, and mounted with Hardset Vectashield plus DAPI mounting medium (Vector Laboratories, Burlingame, CA, USA; H1500) and sealed with coverslips. Images were obtained using Nikon Eclipse TE 200 (Nikon, Melville, NY, USA) and Zeiss LSM 780 microscopes (Zeiss, White Plains, NY, USA). The images were captured with constant parameters and were quantified blind to treatment group. Staining was analyzed by Fiji software (ImageJ, Bethesda, MD, USA) at the same threshold setting for each immunostained marker.

### 2.6. Colocalization of DUSP4 with Neurons, Microglia, or Astrocytes via Confocal Microscopy

To assess whether AAV-DUSP4 overexpression was cell type-specific, NeuN (neuron), IBA1 (microglia), or GFAP (astrocyte) was co-stained with DUSP4 in 30-μm brain sections of each animal. The detail of the staining process was as described above. Colocalization was analysed using the JACoP plugin in ImageJ by a lab technician who was blinded to the treatment groups. The output of colocalization was calculated using thresholded Manders’ correlation coefficient of global statistical analysis, considering pixel intensity distributions. At least 4 brain sections were analysed per animal, and the percentage of the fraction of DUSP4 in the neurons, microglia, or astrocytes was expressed as the mean ± standard error of the mean (SEM).

### 2.7. Quantification of Amyloid Plaque Load

The image quantification of 6 × 10^10^ was performed by lab technicians who were blinded to the treatment groups. First, the images were quantified based on mean gray value of percentage thresholded area in the dHc. Then, the total numbers of amyloid plaque clusters from each brain section were manually counted by technicians. The results are represented in 6 × 10^10^ intensity or number of plaques in dHc.

### 2.8. Aβ Assays

Hippocampal Aβ^1–40^ and Aβ^1–42^ from RIPA-extracted supernatants were quantified by human/rat Aβ^1–40/1–42^ ELISA kits (Fujifilm Wako Chemicals, Richmond, VA, USA; #294-64701, #290-6260) following the manufacturer’s instructions. Absolute concentrations of Aβ were normalized to the initial tissue weight.

### 2.9. Western Blotting

Protein concentrations were determined by Pierce BCA Assay Kit (ThermoFischer Scientific). Equal amounts of protein (20 µg) from each sample were separated by electrophoresis in precast 4–12% Bis-Tris gels (Bio-Rad, Hercules, CA, USA) and transferred to polyvinylidene difluoride membranes using the iBlot system (Invitrogen, Atlanta, GA, USA). Membranes were then incubated in Odyssey blocking buffer for 1 hr at room temperature before incubation with the following primary antibodies in blocking buffer (Odyssey) and 0.1% Tween-20 at 4 °C overnight: anti-DUSP4 (1:1000, Abcam, Cambridge, MA, USA; ab216576); anti-Aβ (1:1000, Biolegend, San Diego, CA, USA; 803001); anti-phospho-Tau (1:1000, Invitrogen, Atlanta, GA, USA; 44-738G); anti-Tau (1:1000, Invitrogen, Atlanta, GA, USA; MN1000); anti-ERK (1:1000, Cell Signaling, Danvers, MA, USA; 4695S); anti-p-ERK (1:1000, Cell Signaling, Danvers, MA, USA; 9101S); rabbit anti-C99 (1:2000, gift from Dr. Nikolaos Robakis); or anti-actin (1:1000, Millipore Sigma, St. Louis, MO, USA; MAB1501). In the second day, membranes were washed with 0.1% Tween-20, and then incubated with a mixture of secondary antibodies: goat anti-rabbit 800CW (1:15,000, LI-COR, Lincoln, NE, USA) and goat anti-mouse 680LT (1:20,000, LI-COR, Lincoln, NE, USA) in Odyssey blocking buffer with 0.01% SDS and 0.1% Tween-20 at room temperature for 1 h. Then the membranes were washed with 0.1% Tween-20 in PBS followed by PBS. The membranes were analyzed using an Odyssey infrared imager (LI-COR, Lincoln, NE, USA). Protein bands were quantified using Odyssey Imager analysis software and were normalized using actin as an internal loading control.

### 2.10. RNA Seq and Differential Expression Analysis

RNAseq on hippocampi from these mice was performed by Novogene using Illumina Novaseq 6000 S4 flow cells. Only the samples with RNA integrity number (RIN) >9 were selected for further analysis. NEB kit was used to construct non-directional libraries. RNAseq assays were performed after ribosomal RNA depletion by Ribo-Zero. Four main steps of data QC were implemented including determination of the (1) distribution of sequencing error rate; (2) distribution of sequencing quality; (3) results of raw data filtering; and (4) distribution of A/T/G/C bases. The filtering process included: (1) removal of reads containing low quality bases (Qscore ≤ 5) that were over 50% of the total bases contained in the read, (2) removal of reads containing adapters, and (3) removal of reads containing N > 10% (N represents bases that cannot be determined). Raw counts of each gene were obtained by aligning RNA reads to mm10 reference genome using STAR [33] (version 2.7.5b). Differential expression was analyzed in R (version 3.6.3) using edgeR [34]. Genes with counts per million reads greater than 1 in at least 5 samples were included for further analysis. Differentially expressed genes with FDR < 0.05 were considered statistically significant in each comparison.

### 2.11. Human Brain Data Analysis Methods

To analyze the expression pattern of *DUSP4*, *VGF* and *BDNF* genes in human AD brains, we first used our recently published RNAseq data in the parahippocampal gyrus (PHG) region of postmortem brains from AD patients and controls from the Mount Sinai Brain Bank (MSBB) [35,36]. Here we used the preprocessed data that had been normalized and corrected for known covariates, except sex which was excluded from the covariate correction. Using boxplots, we visualized the sex-specific gene expression distribution stratified by CDR. Meanwhile, we calculated the Spearman correlation coefficients between gene expression values and CDR scale in males and females separately.

### 2.12. Analysis of the University of California, Irvine (UCI) 5xFAD Mouse RNAseq Data

We downloaded the raw RNAseq data from the hippocampi of 4-, 12- and 18-month-old 5xFAD mice from the AMP-AD portal, synapse.org [37]. 5xFAD transgenic mice on a congentic C57BL/6J genetic background overexpress both mutant human PSEN1 harboring two FAD mutations, L286V and M146L, as well as human amyloid beta (A4) precursor protein 695 (APP) with the Florida (I716V), Swedish (K670N, M671L), and London (V717I) familial AD (FAD) mutations. Details about the sample processing, library construction and sequencing are also available at synapse.org [37].

Paired-end 43bp sequencing reads were aligned to mouse reference genome mm10 using STAR aligner v2.5.3a [33] guided by a customized mouse GENCODE gene model release v15. Mapped reads were summarized to gene levels using the featureCounts program v1.6.3 [38]. Raw count data were normalized as counts per million at the log2 scale by the voom function in the R limma package [39]. Expression patterns of selected genes *Dusp4*, *Vgf*, and *Bdnf* were visualized per age-genotype-sex combinations by boxplot. The expression difference between 5xFAD and age-sex-matched control was calculated by one-sided *t*-test.

### 2.13. Derivation of Microglial Cells from hiPSCs

All human induced pluripotent stem cell (hiPSCs) lines with a normal karyotype were regularly checked and confirmed negative for mycoplasma. The hiPSCs maintained in mTeSR1 medium according to WiCell protocols were dissociated by Accutase to obtain a single-cell suspension. In total, 10,000 cells were plated in each well of an ultra-low-attachment 96-well plate (Corning, St. Louis, MO, USA) in mTeSR1 medium with 50 ng/mL of human BMP4, 50 ng/mL of human VEGF, 20 ng/mL of human SCF, and 10 μM Rho-associated protein kinase (ROCK) inhibitor. Embryoid bodies were fed every day from day 1 to day 3, then transferred to 6-well plates (Corning, St. Louis, MO, USA) in the differentiation media containing X-VIVO 15 media, 2 mM GlutaMAX, 50 U/mL of penicillin-streptomycin, 0.055 mM 2-mercaptoethanol, and supplemented with 50 ng/mL of human SCF, 50 ng/mL of human M-CSF, 50 ng/mL of human IL3, 50 ng/mL of human FLT3 and 5 ng/mL of human TPO. After 4 days, embryoid bodies were fed with the same differentiation media. On day 11, a medium change was performed and embryoid bodies were maintained in differentiation media plus 50 ng/mL of human FLT3, 50 ng/mL of human M-CSF and 25 ng/mL of human GM-CSF. On day 18, floating microglial progenitors in the medium were collected and cultured in RPMI 1640 medium containing 10 ng/mL GM-CSF and 100 ng/mL IL-34 for 2 weeks.

### 2.14. Cortical Neuron Differentiation from hiPSCs

The hiPSCs were dissociated with Accutase and plated at 200,000 cells per cm^2^ onto Matrigel-coated plates in mTeSR1 medium with ROCK inhibitor (10 μM). After 1 to 2 days cells were treated with differentiation media (DMEM/F12:Neurobasal (1:1), 2 mM GlutaMAX, 1% N2 supplement, 2% B27 minus Vitamin A supplement) containing LDN193189 (100 nM), SB431542 (10 μM) and XAV939 (1 μM) for 10 days of differentiation. Cultures were fed with differentiation media with XAV939 (1 μM) for an additional week to allow the expansion of neural progenitor cells. Neural progenitor cells were then dissociated and replated on poly-l-ornithine/fibronectin/laminin-coated plates and maintained in BrainPhys Basal medium containing B-27 supplement, BDNF (40 ng/mL), GDNF (40 ng/mL), laminin (1 μg/mL), ascorbic acid (200 μM), and dbcAMP (250 μM), for neuronal differentiation and maturation, with the addition of SU5402 (10 μM), DAPT (10 μM) and PD0325901 (10 μM) for the first week of differentiation.

### 2.15. RNA In Situ Hybridization

Brain sections (30 μm) were used in RNA in situ hybridization (RNAscope^®^). RNAscope^®^ fluorescent in situ hybridization was performed according to the manufacturer’s instructions (Advanced Cell Diagnostics, Inc., Newark, CA, USA). Briefly, the mounted sections were washed in PBS followed by incubation of the sections at 60 °C for 30 min (Lab-Line Instruments Inc., Melrose Park, IL, USA). Then the mount sections were postfixed in 4% PFA at room temperature (RT) for 1 h. After fixation, the sections were dehydrated in ethanol by incubating in 50%, then 70%, then 100% ethanol for 5 min and then the sections were allowed to dry at RT for 45 min. Then the sections were incubated with H_2_O_2_ for 10 min and washed with distilled water. Next, target retrieval step was performed by incubating the slides in a boiling retrieval reagent for 5 min. Slides were then washed in distilled water followed by soaking in 100% ethanol, and finally the sections were allowed to air dry for 5 min at RT. Then the sections were incubated with protease III at 40 °C in the pre-warmed ACD HybEZ II Hybridization System (Cat. No. 321721, ACD) for 20 min inside the HybEZ Humidity Control Tray (Cat. No. 310012, ACD). Sections were then washed twice with distilled water. The following probes from ACD were used and diluted at 1:50 (Rbfox3-C4:mouse-Dusp4): Mouse *Dusp4* (NM_176933.4, bp431-1523, Cat. No. 425871), and *Rbfox3*-C4 (NM_001039167.1, bp1827-3068, Cat. No. 313311-C4). Then the sections were incubated with a corresponding probe at 40 °C for 2 h in the HybEZ oven (ACD), washed twice with 1× wash buffer, and stored overnight at RT in 5x SSC buffer (Thermo Fisher Scientific, Fair Lawn, NJ, USA). The next day, slides were washed twice with wash buffer (2 min per wash), followed by the three amplification steps (AMP 1, AMP 2, and AMP 3 at 40 °C for 30, 30, and 15 min, respectively, and the samples were washed twice with wash buffer after each amplification step). Then the sections were treated with the HRP reagent corresponding to each probe channel (e.g., HRP-C1 or HRP-C4) at 40 °C for 15 min, followed by the TSA Plus fluorophore Opal 520 or 620 (dilution of 1:750 in RNAscope^®^ Multiplex TSA Buffer [Cat. No 322809]) at 40 °C for 30 min, and HRP blocker at 40 °C for 15 min, with two wash steps after each of the incubation steps. Then the microglial cells were detected by using rabbit anti-IBA1 antibody (Fujifilm Wako Chemicals, Richmond, VA, USA; Cat. No 01919741) and secondary Alexa Fluor 488 anti-rabbit antibody. Finally, the slides were counterstained with DAPI for 30 s. Images were obtained using Zeiss LSM 780 microscopes.

### 2.16. Statistics

All data, except genomic data, were analyzed using GraphPad Prism. Graphs are represented in mean of each group ± SEM. Sample sizes (*n* values) and statistical tests are shown in the figure legends. One-way or two-way ANOVA was performed for multiple comparisons. A Student’s *t*-test was used for simple comparisons. Significance is indicated with * *p* < 0.05, ** *p* < 0.01, *** *p* < 0.001 and **** *p* < 0.0001.

## 3. Results

### 3.1. Downregulation of DUSP4, BDNF and VGF in Human AD and 5xFAD Hippocampus

*DUSP4* is predicted to be downstream of *VGF* in the *VGF* multiscale gene network (Figure 1A) [14], and expression of the network driver *VGF* is reduced in human AD subjects and in 5xFAD mouse hippocampi [14]. To determine the sex-specific expression patterns in human AD brains of several key hub genes in the *VGF* multiscale gene network, including *DUSP4*, *BDNF* and *VGF*, we used our recently published RNAseq data in the parahippocampal gyrus (PHG) region of postmortem brains of AD subjects and controls from the Mount Sinai Brain Bank (MSBB). The Spearman correlation coefficients show that the downregulation of *DUSP4*, *BDNF* and *VGF* gene expression is correlated with the clinical dementia rating (CDR) in both sexes, with the exception that *DUSP4* gene expression in female AD brains has no significant correlation with CDR (Figure 1B–D).

Our validation studies reported here and previously [26] utilize 5xFAD on a mixed B6/SJL genetic background, but the availability of raw RNAseq data (AMP-AD portal) from 5xFAD mice on a C57BL/6J genetic background allowed us to assess hippocampal *Dusp4*, *Vgf* and *Bdnf* gene expression at 4, 12, and 18 months of age on a congenic background (Figure 1E–G). At 4 months of age, female 5xFAD mice showed a significant decrease in hippocampal *Dusp4*, *Vgf* and *Bdnf* expression (Figure 1E–G), consistent with the well-documented increased severity of amyloid neuropathology at this age in females compared to males [40]. Although the patterns of gene expression in Figure 1E–G are complex, expression of each of the three genes was significantly reduced in 4- to 18-month-old male and female 5xFAD mice, with peak reductions in hippocampal *Bdnf* gene expression noted in 12-month-old female and male 5xFAD mice compared to age- and sex-matched WT mice.

### 3.2. Overexpression of DUSP4 in Dorsal Hippocampus Improves Spatial Learning Behavior in 5xFAD Mice in a Sex-Dependent Manner

We measured endogenous *Dusp4* mRNA levels in 5-month-old 5xFAD mice on a mixed B6/SJL genetic background. We observed a significant reduction of hippocampal *Dusp4* mRNA levels in female 5xFAD-GFP mice [5xFAD following injection of AAV5-GFP into the dorsal hippocampus (dHc)], while *Dusp4* mRNA levels in male 5xFAD-GFP mice were significantly increased compared to WT-GFP (Figure 2A,B). To determine whether overexpression of DUSP4 would ameliorate AD-associated neuropathology as demonstrated following overexpression of the network driver VGF [14], we injected AAV5-DUSP4 or AAV5-GFP (control) into the dHc of 4-month-old male and female WT or 5xFAD mice (Figure 2C). To confirm the overexpression, hippocampal mRNA levels were quantified by RT-PCR, and *Dusp4* mRNA was robustly increased one month after injection, compared to the AAV5-GFP-injected control (Figure 2D, left). Western blot analyses confirmed overexpression of hippocampal DUSP4 protein (Figure 2D, right; Appendix A).

We utilized bioinformatic tools, immunohistochemical analysis, RNAscope, and RT-PCR to determine which cell type(s) in the brain endogenously express and/or overexpress DUSP4 after AAV injection. As per the brainrnaseq.org single cell RNAseq database, mouse *Dusp4* mRNA is predominantly expressed in neurons, with lower levels in microglia, astrocytes, and endothelial cells (Appendix A). Relatively low expression levels of human *DUSP4* mRNA are detected by single cell RNAseq in excitatory and inhibitory neurons, with no detectable expression in astrocytes and microglia (Appendix A). To confirm cell type(s) expressing *DUSP4* in human samples, we determined *DUSP4* mRNA levels in human iPSC-derived microglia and neurons. RT-PCR indicated that *DUSP4* mRNA levels were significantly higher in neurons than in microglia (Appendix A). We then examined the cellular distribution of overexpressed DUSP4 after AAV5-DUSP4 administration. Hippocampal sections were immunohistochemically co-stained with anti-DUSP4, anti-IBA1 (marker for microglia), anti-GFAP (marker for astrocytes) and anti-NeuN (marker for neuronal nuclei) (Figure 2E–H). The staining images at lower magnification revealed that DUSP4 protein was mainly localized in the pyramidal neuron layer of the hippocampus (Figure 2E). The magnified images, covering part of the hilus and dentate gyrus (DG) region, showed that DUSP4 co-localized almost exclusively with NeuN (Figure 2F), but not with IBA1 (Figure 2G) or GFAP (Figure 2H). Quantification confirmed that DUSP4 was mainly overexpressed in neurons (Figure 2I), suggesting that AAV5-mediated overexpression of DUSP4 was primarily targeted to neurons. Moreover, staining of hippocampal *Rbfox3* mRNA (neuronal marker, Figure 2J top left), *Dusp4* mRNA (Figure 2J middle left), and merged image (Figure 2J bottom left) by RNAscope also showed the localization of *Dusp4* mRNA in the pyramidal neuron layer, while there was no detectable *Dusp4* mRNA in IBA1-positive cells (Figure 2J right panel). Together, these data indicated that AAV5-mediated overexpression of DUSP4 was localized primarily in dHc neurons.

To determine whether DUSP4 overexpression improved learning behavior, we used the Barnes maze test to assess male and female mice overexpressing DUSP4 or GFP in the dHc at 5 months of age. Consistent with a learning behavior deficit, male 5xFAD-GFP took more time to enter the hidden tunnel (escape box) (Figure 3A) and traveled less distance in the target quadrant (location of escape box) (Figure 3B), compared to male WT-GFP, in the 5-day training session. Barnes maze behavior of male 5xFAD-DUSP4 and 5xFAD-GFP mice were indistinguishable (Figure 3A,B) except on day 5, when male 5xFAD-DUSP4 exhibited a trend of increase in the percentage of distance traveled in the target quadrant (Figure 3B), perhaps suggesting that these mice may have experienced a subtle improvement in learning behavior. Like male 5xFAD-GFP, female 5xFAD-GFP took more time to enter the hidden tunnel (Figure 3C) and traveled less distance in the target quadrant (Figure 3D), compared to female WT-GFP. Female 5xFAD-DUSP4 mice exhibited a reduction in time to enter the hidden tunnel compared to female 5xFAD-GFP, and their learning behavior was indistinguishable from that of female WT-GFP (Figure 3C). In addition, female 5xFAD-DUSP4 mice also showed a significant increase in the percentage of distance traveled in the target quadrant compared to 5xFAD-GFP (Figure 3D), indicating that DUSP4 overexpression rescued the learning behavior deficit in female 5xFAD mice.

### 3.3. Overexpression of DUSP4 Reduces Amyloid Plaque Load, Aβ^1–42^ and Aβ^1–40^ Levels, and Phospho-Tau Levels in Male and Female 5xFAD Mice

Amyloid plaques and hyperphosphorylation of Tau are hallmarks of AD, and one potential mechanism underlying accumulation of aggregates involves dysregulation of APP processing that can lead to overproduction of Aβ. We therefore investigated whether DUSP4 overexpression reduced amyloid burden in 5xFAD mice. There was a reduction in amyloid plaque burden as determined by staining with 6E10, in the hippocampi of both female and male 5xFAD-DUSP4 mice compared to 5xFAD-GFP (Figure 4A,B). Western blot analysis of either RIPA or 8M urea hippocampal extracts showed that there were no changes in holo-APP levels in female or male 5xFAD mice overexpressing DUSP4 compared to GFP (Figure 4C,D; Appendix A). On the other hand, ELISA indicated that hippocampal Aβ^1–42^ and Aβ^1–40^ levels were reduced in both female and male 5xFAD-DUSP4 compared to 5xFAD-GFP mice, in either RIPA (Figure 4E) or 8M urea (Figure 4F) hippocampal lysates, suggesting that overexpression of DUSP4 may regulate APP processing in 5xFAD mice. We further determined by western blot analysis that DUSP4-overexpressing male and female 5xFAD mice also had a marked decrease in APP C-terminal fragment (APP-CTF) C99 levels, which is generated by β-secretase cleavage of APP, compared to GFP-overexpressing 5xFAD mice (Figure 4G, female RIPA extract, and Figure 4H, female and male 8M urea extracts), while the C99 protein fragment was not detected in RIPA extracts of either DUSP4-overexpressing or GFP-overexpressing male 5xFAD mice. In addition, we also quantified phosphorylated Tau (Thr205) levels, finding that DUSP4 overexpression significantly decreased p-Tau levels in 5xFAD female and male mice (Appendix A).

### 3.4. DUSP4 Overexpression Does Not Regulate APP Secretase-Processing Enzyme Gene Expression in 5xFAD Mice

To explore the potential mechanisms by which DUSP4 overexpression altered Aβ accumulation in 5xFAD mice, we measured levels of three major APP secretase-processing enzyme RNAs. Activities of three key enzymes, α, β and γ-secretases, regulate APP processing, and Aβ peptide is produced by the proteolytic action of β and γ-secretases on APP in the amyloidogenic pathway. We quantified the expression levels of mouse γ-secretase subunit gene transcripts for *Psen1*, *Psen2* and the human *PSEN1* transgene, and the β-secretase component gene *Bace1* in 5xFAD mice by RT-PCR. Expression of most of these genes was upregulated in both female and male 5xFAD-GFP, with the exception of *Psen2* mRNA levels in male and *Bace1* mRNA levels in female, where no changes were observed when compared to WT-GFP mice (Appendix A). Transcripts encoded by γ-secretase subunit genes including *Nct*, *Aph1a*, *Aph1b*, and *Aph1c* were quantified, and no changes were observed in 5xFAD mice overexpressing DUSP4 compared to 5xFAD-GFP (Appendix A). Human *PSEN1* mRNA levels, transcribed from the Thy1 promoter-driven *APP-PSEN1* transgene in 5xFAD, were significantly downregulated in both female and male 5xFAD-DUSP4 compared to 5xFAD-GFP mice (Appendix A) while the levels of endogenous *Psen1*, *Psen2*, and *Bace1* mRNAs were not affected (Appendix A).

To investigate whether DUSP4 overexpression regulated the non-amyloidogenic APP processing pathway, we quantified α-secretases *Adam10* and *Adam17* mRNA levels by RT-PCR. Levels of *Adam10* and *Adam17* mRNAs in 5xFAD were previously reported as increased at one month of age, and by nine months of age, *Adam10* mRNA was decreased and *Adam17* also showed a trend of reduction at the same age [41]. We found that *Adam10* and *Adam17* mRNAs were actually upregulated in female and male 5xFAD-GFP at 5 months of age compared to WT-GFP (Appendix A). Overexpression of DUSP4 in either female or male 5xFAD mice did not affect the levels of *Adam10* and *Adam17* mRNAs compared to 5xFAD-GFP. Taken together, these results indicate that the amelioration of amyloid burden induced by DUSP4 overexpression is not obviously caused by widespread changes in the expression of any component of the major APP secretase processing enzymes.

### 3.5. DUSP4 Overexpression Downregulates Vgf-Associated Network Genes

In anticipation of investigating whether DUSP4 overexpression modulated the VGF network, we first determined whether *Vgf*-associated genes are dysregulated in 5-month-old 5xFAD mice by comparing hippocampal *Vgf*, *Sst*, *Bdnf*, and *Scg2* mRNA levels in 5xFAD-GFP and WT-GFP mice (Appendix A). We observed a reduction of *Vgf*, *Sst*, and *Scg2* mRNA levels in female 5xFAD-GFP mice compared to WT-GFP, but no change in male 5xFAD-GFP mice (Appendix A). Female 5xFAD-GFP also showed a trend toward reduction in *Bdnf* mRNA levels, but no change was observed in male 5xFAD-GFP mice (Appendix A). These sexually dimorphic differences in gene expression may be due, at least in part, to the well-documented increased severity of amyloid pathology in female 5xFAD at this age relative to male [40].

To test the predictions of the multiscale VGF network (Figure 1A), we determined the effect of DUSP4 overexpression on DUSP6 expression levels. First, we determined DUSP6 expression levels in 5xFAD mice by comparing the hippocampal *Dusp6* mRNA and DUSP6 protein levels in 5xFAD-GFP and WT-GFP mice. We observed an upregulation of *Dusp6* mRNA levels, but not protein levels, in both female and male 5xFAD-GFP (Appendix A), suggesting a dysregulation of DUSP6 at the mRNA level in 5xFAD mice. Curiously, overexpression of DUSP4 downregulated *Dusp6* mRNA and DUSP6 protein levels in both female and male 5xFAD-DUSP4 and WT-DUSP4 mice compared to the control (Appendix A). Appendix A shows the summary of expression changes in VGF network-associated genes caused by DUSP4 overexpression in 5xFAD. These data indicate that DUSP4 has regulatory effects on *Vgf*-associated genes, which may contribute to improvement of the learning behavior and reduction of amyloid burden in 5xFAD mice.

### 3.6. Neuroinflammatory, ERK/MAPK, and Interferon Signaling Pathway Gene Expression, Which Is Upregulated in Female 5xFAD Hippocampus Relative to Wild Type, Is Downregulated by DUSP4 Overexpression

To determine the molecular pathways in 5-month-old 5xFAD mice that are affected by DUSP4 overexpression, we used bulk RNAseq to generate transcriptomic profiles from the dHc of both female and male WT overexpressing GFP or DUSP4. There were 93 differentially expressed genes (DEGs) in female WT-DUSP4 relative to female WT-GFP (FDR < 0.05) (Appendix A). Upregulated DEGs included alpha-2-macroglobulin (*A2m*) [42], syndecan 1 (*Sdc1*) [43], and interferon-induced transmembrane protein 2 (*Ifitm2*) [44], and these genes have either been directly or indirectly associated with the metabolism of Aβ. To identify pathways affected by DUSP4 overexpression in female WT, DEGs were analyzed by pathway enrichment analysis. Consistent with the known roles for DUSPs in regulating ERK/MAPK activity, pathway enrichment analysis of this set of DEGs highlighted ERK/MAPK signaling (Appendix A). Conversely, there were only 5 DEGs observed in male WT mice overexpressing DUSP4 (FDR < 0.05) (Appendix A). To investigate whether there was an effect of chronic DUSP4 overexpression on ERK/MAPK activation, phosphorylated ERK levels were determined by western blot analysis, with no change noted in p-ERK (pThr202, pTyr204) levels in DUSP4-overexpressing female and male 5xFAD mice (Appendix A).

We then assessed transcriptomics in 5xFAD mice overexpressing DUSP4 or GFP in dHc, compared to 5xFAD-GFP or WT-GFP, respectively. We identified 2850 DEGs in female 5xFAD-GFP compared to WT-GFP, most of which were upregulated (FDR < 0.05) (Figure 5A). Comparison of female 5xFAD-DUSP4 to 5xFAD-GFP revealed 202 downregulated DEGs (FDR < 0.05) (Figure 5B). Enrichr pathway enrichment analysis showed, as previously described [45,46,47], that many inflammatory pathways in the hippocampi of female 5xFAD were upregulated compared to WT (Figure 5C). Overexpression of DUSP4 in female 5xFAD downregulated these inflammatory pathways (Figure 5D). Comparison of the DEGs from female 5xFAD-DUSP4 vs. 5xFAD-GFP to 5xFAD-GFP vs. WT-GFP revealed 194 DEGs in common (Figure 5E), and most of these DEGs are associated with inflammatory pathways, including *Ccl12*, *Ccl2*, *Irf1*, and *Aif1*. Ingenuity Pathway Analysis (IPA) also predicted regulation of similar pathways, including downregulation of ERK/MAPK, interferon and neuroinflammatory pathways, by DUSP4 overexpression in female 5xFAD (Figure 5F). The predicted DEGs in each of these pathways are listed in Figure 5G, as are several genes in the PD-L1/PD-1 immune checkpoint pathway, that are reduced in expression by DUSP4. To confirm the changes of gene expression by DUSP4 overexpression, selected genes from both ERK/MAPK and neuroinflammatory pathways were analyzed by RT-PCR. The results showed that these genes were upregulated in female 5xFAD mice, and all were downregulated by DUSP4 overexpression, consistent with the data from the RNAseq analysis (Figure 5H). Conversely, there were no DEGs observed in male 5xFAD-GFP compared to WT-GFP, consistent with previous reports that female 5xFAD develop more severe neuropathology than age-matched males [48,49], while 2 DEGs were found in male 5xFAD-DUSP4 when compared to male 5xFAD-GFP (Appendix A). Lastly, we investigated whether DUSP4 overexpression affected AD-associated microgliosis and microglial activation, finding that DUSP4 overexpression reduced Iba1 and Cd68 mRNA levels and IBA1 immunohistochemical fluorescence intensity, in the dorsal hippocampus, in female but not male 5xFAD mice (Appendix A).

## 4. Discussion

Our study implicates DUSP4 function in the pathogenesis and progression of AD-related phenotypes in the 5xFAD mouse model. DUSP4 overexpression in dHc rescued deficits in learning behavior in a sexually dimorphic manner, in female but not male 5xFAD, despite comparable amelioration of amyloid burden in mice of both sexes. Utilizing bioinformatics, and DUSP4 protein and mRNA localization, we further demonstrated that in mice, DUSP4 is primarily expressed in neurons with lower levels in microglia, astrocytes, and endothelial cells; in human brains, expression levels are also highest in neurons. A caveat of our experimental approach utilizing intrahippocampal AAV-5 administration, which was confirmed in our localization of overexpressed DUSP4 mRNA and protein (Figure 2), is that AAV5-DUSP4 is primarily transduced in neurons, so our experiments therefore assessed the functional impact of DUSP4 modulation in neurons.

MAPKs play a crucial role in cellular responses to cytokines and external stress by regulating the production of inflammatory mediators [50]. As a modulator of MAPK phosphorylation and signal transduction, DUSP4 is a potential regulator of neuroinflammation, including in AD. Studies using DUSP4 knockout mice have identified roles for this protein in inflammatory and anti-microbial immune responses through the regulation of MAPK signaling [51]. Consistent with these findings, our Ingenuity Pathway Analysis (IPA) indicated that DUSP4 overexpression downregulated neuroinflammatory and ERK/MAPK signaling pathways in the female 5xFAD hippocampus.

Interestingly, IPA identified type-II interferon gamma (IFNγ) proinflammatory signaling as the most robustly downregulated pathway by DUSP4 overexpression in the female 5xFAD hippocampus (Figure 5D). Previous temporal gene profiling in the 5xFAD hippocampus and cortex revealed numerous overexpressed IFNγ-regulated transcripts, including those that encode chemokines, major histocompatibility complex (MHC) class I and II molecules, and regulators of the IFNγ pathway, autophagy and phagocytosis [47], which mirrors gene expression profiling in AD subjects [52,53]. IFNγ levels have not been found to be elevated in AD, but expression levels are higher in several transgenic mouse models of AD [54,55,56,57]. IFNγ is thought to be primarily synthesized and secreted by Th1 T cells that infiltrate the brain in patients with AD and in mouse models of AD, escalating neuroinflammatory cascades [58,59,60,61,62]. Reports, however, demonstrate varied effects of administered IFNγ on AD progression in mouse models [63] including: (1) improved cognition and Aβ clearance in 8-month-old APP/PS1 mice following intraperitoneal IFNγ infusion [64], (2) reduced phospho-tau pathology and increased neurogenesis and amyloid pathology in 12-month-old 3xTG-AD mice after intra-hippocampal AAV1-IFNγ administration [65], (3) increased Aβ deposition in 6-month-old APP/PS1 following intra-hippocampal administration of AAV8-IFNγ [66], and (4) attenuated amyloid deposition following either neonatal AAV1-IFNγ intraventricular infusion or intra-hippocampal IFNγ administration in 5-month-old APP TgCRND8 mice [67].

In 5xFAD mice, ERK positively regulates disease-associated microglia (DAM) gene expression, including *Ccl3*, *Ch25h*, *Spp1*, *Igf1*, and *Itgax*, and pro-inflammatory DAM genes, including *Ccl2*, *Irf1*, *Tnf*, *Slamf9*, and *Cd69* [68]. Our RT-qPCR and immunohistochemistry results also indicated a marked decrease of microglial markers in DUSP4-overexpressing female 5xFAD mice, but not male 5xFAD mice (Appendix A). Inhibition of ERK activation in primary microglia mitigates IFNγ-induced effects, pro-inflammatory gene expression, and neuronal phagocytic activity [68]. Consistent with these data, our RNAseq analysis showed upregulation of these DAM genes in 5xFAD mice (Figure 5A). Intriguingly, DUSP4 overexpression in 5xFAD downregulated some of these pro-inflammatory DAM genes, including *Ccl2*, and *Irf1* (Figure 5B), and also downregulated interferon signaling, TYROBP network hubs and drivers, and ERK/MAPK signaling (Figure 5D,F). However, we did not detect any significant change of p-ERK levels in DUSP4-overexpressing WT and 5xFAD mice compared to GFP overexpression in our WT and 5xFAD groups, respectively (Appendix A). We hypothesize the chronic exposure to DUSP4 overexpression may have resulted in compensatory, homeostatic changes in p-ERK levels. Nevertheless, in microglia, it has been demonstrated that p-ERK levels had an inverse relationship with DUSP4 levels within hours of lipopolysaccharide exposure [69], which further supports that DUSP4 acutely regulates p-ERK levels. These results are consistent with DUSP4 overexpression reducing pro-inflammatory microglial responses by regulating the ERK/MAPK pathway. In accord with our transcriptomics analysis, preliminary proteomics and phospho-proteomics analysis of the hippocampus from DUSP4-overexpressing 5xFAD mice has also identified downregulation of IFNγ and MAPK/JNK signaling pathways (E. Wang, A.L. Pan et al., manuscript in preparation).

A recent study has further identified a critical role for IFNγ pathways and IFN-induced transmembrane protein 3 in the regulation of γ-secretase activity and amyloid plaque deposition in AD [70]. We did not, however, detect any evidence that APP secretase-processing enzyme mRNA levels were altered by DUSP4 overexpression, but γ-secretase catalytic activity was not directly assayed in our studies. In addition to type-II interferon signaling, type-I interferon pathways are upregulated in AD patients and are significantly correlated with disease severity, and in mouse AD models, type-I signaling modulates microglial activation and complements C3-dependent synapse elimination [71]. Ablation of type-I interferon signaling is neuroprotective in APP/PS1 mice [72], and this pathway has been implicated in Aβ-induced microgliosis and neuroinflammation, with type-I interferon having been reported to inhibit microglial phagocytosis [73]. Reduced interferon signaling in DUSP4-overexpressing female 5xFAD (Figure 5F) would therefore be anticipated to be neuroprotective and to reduce microgliosis and neuroinflammation.

Based on our analysis, neurons are the predominant cell type that overexpress AAV5-encoded DUSP4. Therefore, suppression of inflammatory pathways in microglia via DUSP4 overexpression in neurons would likely require non-cell autonomous, neuronal–microglial crosstalk, as depicted in Figure 5I, although the mechanism(s) and pathways underlying this crosstalk remain speculative. A number of signaling molecules with microglial receptors have been proposed to mediate crosstalk, including purine derivatives of ATP (e.g., adenosine), classical complement cascade factors (e.g., C1q, C3, C3a), neurotransmitters (e.g., GABA, glutamate), cytokines, chemokines (e.g., CX3CL1/fractaline), and neurotrophins, many of which are released in response to synaptic activity [74,75,76,77,78,79]. We found that DUSP4 overexpression downregulated *Bdnf* mRNA levels in female 5xFAD, encoding the neurotrophin BDNF, a potential mediator of neuronal-microglial crosstalk. Another possible underlying mechanism is that DUSP4 overexpression modulates IFNγ/JAK/STAT signaling in neurons [62,63], leading to alterations in the expression of secreted cytokines, chemokines, and/or growth factors that could be involved in neuronal-microglial crosstalk. Lastly, miR-124-containing exosomes, secreted by APP Swedish SH-SY5Y (SWE) neurons, influence microglial activation by IFNγ and IFNγ-induced inflammatory proteomic signatures, with potential relevance to AD-associated neurodegeneration [80].

In addition, our IPA showed several DEGs associated with the PD-L1/PD-1 pathway, including *Cd274* (which encodes the PD-L1 ligand), were upregulated in female 5xFAD compared to WT and downregulated by DUSP4 overexpression in female 5xFAD (Figure 5G). The PD-L1/PD-1 pathway is an immune checkpoint with an important role in maintaining immune homeostasis by tuning the immune response and regulating apoptosis and cytokine secretion. PD-L1 levels are increased in the cerebrospinal fluid of AD patients, and expression of PD-L1 and PD-1 in astrocytes and microglia, respectively, was upregulated in proximity to amyloid plaques in AD patients and APP/PS1 mice [81], consistent with PD-L1 and PD-1 being expressed under conditions of chronic inflammation and a role for this pathway in the microglial phagocytosis of amyloid. We found that hippocampal expression of *Cd274* (encoding PD-L1) was reduced by DUSP4 overexpression, although it is not clear whether DUSP4 overexpression directly reduced PD-L1 levels or whether this was an indirect response to the reduction in amyloid load by DUSP4 overexpression. Studies have shown that the exposure of human neuronal cells to an inflammatory cytokine such as Interleukin-18 (IL-18) or a combination of INFγ and TNFα increases Aβ production [82,83], so it is also possible that reduced amyloid load is the consequence of mitigated neuroinflammation due to DUSP4 overexpression.

Hippocampal overexpression of VGF has a protective effect in 5xFAD, reducing both Aβ accumulation and memory deficits [14]. Because VGF and DUSP4 are located within the same network, we hypothesized that DUSP4 overexpression might reduce Aβ toxicity by upregulating the expression of *Vgf* and other genes in the network (*Sst*, *Bdnf*, *Scg2*, and *Dusp6*). Somewhat surprisingly, we found that *Vgf*, *Sst*, *Bdnf*, *Scg2*, and *Dusp6* were downregulated by DUSP4 overexpression. Each of these genes is transcriptionally regulated by the cAMP response element-binding protein (CREB) [84]. CREB activation/phosphorylation is reduced by ERK inactivation [85], which would be driven by increased DUSP4 expression, providing a potential mechanism underlying reduced expression of these genes.

We also identified sexual dimorphism in the rescue of Barnes maze learning behavior impairment in 5xFAD mice following DUSP4 overexpression. Amyloid plaque deposition in 5xFAD mice occurs at younger ages and more robustly in females [49,86]. In addition, levels of hemi-brain chemokine (*Ccl2* and *Cxcl10*), cytokine (*Il-1b*, *Tnf-a*, *Il-6*), and glial marker (*Gfap* and *Iba1*) mRNAs were increased in female mice [45]. Thus, the 5xFAD mouse model has significant sexual dimorphism and reduction of neuroinflammatory gene expression is more robust in the DUSP4 overexpressing female 5xFAD hippocampus than in males, at 5 months of age. Male 5xFAD overexpressing DUSP4 were found to have similar reductions in amyloid plaque density as did females, but Barnes maze learning behavior deficits were only rescued in female 5xFAD. We further found that hippocampal Aβ^1–42^ and Aβ^1–40^ levels were reduced in both male and female 5xFAD overexpressing DUSP4, utilizing either RIPA or 8M urea lysis, and that levels of hippocampal APP C-terminal fragment (APP-CTF) C99 were also reduced in 8M urea extracts of male and female 5xFAD overexpressing DUSP4, but were undetectable in RIPA extracts of male 5xFAD. Studies indicate that the β-secretase derived C99 fragment is toxic, and its accumulation may also trigger synaptic and cognitive alterations [87], although reduced C99 levels were found in 8M urea extracts of DUSP4 overexpressing male and female 5xFAD hippocampus, with rescue of cognitive behavior only in females.

In the amyloidogenic pathway, APP is first cleaved by β-secretase to generate either C89 or C99 fragments (depending on the cleavage site), which is followed by γ-secretase cleavage to produce nonpathological Aβ peptide (Aβ^11–40^) from C89, or pathological Aβ peptides (Aβ^1–42^ and Aβ^1–40^) from C99 [87]. Our studies did not show any effects of DUSP4 overexpression in 5xFAD mice on the levels of APP-processing enzyme mRNAs, including those encoding β-secretase (BACE1) and γ-secretase (Appendix A), but we cannot rule out the possibility that DUSP4 overexpression modulated APP-processing enzyme catalytic activities, which would require further investigation. Levels of the C99 fragment are increased by many APP mutations including the Swedish mutations (K670N, M671L) [88,89], which are found in the 5xFAD mouse model, and the C99 fragment is processed to yield amyloidogenic Aβ^1–42^ and Aβ^1–40^ [90,91]. Each of these APP-derived fragments/peptides were found to be decreased by DUSP4 overexpression in our studies. It is also important to note that the current studies were limited to the assessment of RIPA- or 8M urea-soluble amyloid levels, and that additional experimental cohorts and studies utilizing sequential soluble lysis (e.g., TBS and/or RIPA) followed by extraction of the insoluble pellet (e.g., with chaotropic agents and/or formic acid) would be required to determine whether there is any sexual dimorphism in the levels of insoluble Aβ^1–42^ and Aβ^1–40^ between male and female 5xFAD overexpressing DUSP4. Lastly p-Tau levels were reduced in both male and female 5xFAD overexpressing DUSP4.

Dissociation of cognitive outcomes from neuropathology is not uncommon [92] and may be clinically relevant. To date, anti-amyloid pharmacotherapeutics that reduce fibrillar amyloid load are not associated with any obvious meaningful improvement in cognitive deficits [93]. Differences in behavioral phenotypes between male and female 5xFAD overexpressing DUSP4 in our experiments appear to be independent of reduced toxic or amyloidogenic cleavage fragments. Rescue of Barnes maze learning behavior in female 5xFAD therefore likely results from mechanisms that are independent of amyloid plaque deposition, perhaps involving sex-dependent reductions in neuroinflammatory pathway activity [94,95].

There are two caveats regarding the data presented herein. First, using the same cohorts, we wanted to correlate changes in behavior with alterations in neuropathology, amyloid load, and gene expression. Therefore, we cannot formally exclude that a subset of the changes we observed, particularly in gene expression, may have resulted from the behavioral training of AAV-DUSP4-treated mice. However, we think this is unlikely, as the transcriptomic changes we observed in female 5xFAD-DUSP4 mice did not occur in control 5xFAD-GFP or wild-type animals that underwent the same Barnes maze training, and in addition, no DEGs were observed in male 5xFAD mice. Second, the lack of DEGs in male mice prevents us from performing a deeper analysis of the molecular mechanisms underlying sexual dimorphism at this age.

## 5. Conclusions

It has long been known that women are at greater risk of developing Alzheimer’s disease [96], and this includes a greater risk incurred when carrying an ApoE4 allele. In fact, women who are heterozygous for this allele are at equal risk to homozygous men [97]. Highly relevant to our study, more recent data has highlighted the sex-dependent differences in microglial number and function during development and aging [98]. This field of inquiry has progressed to implicate gender differences in microglia in the pathophysiology of AD in humans and in mouse models of tauopathy and amyloidosis. Specifically, there are sex-dependent differences in the microglial miRNA and transcriptome response to Tau abnormalities [99]. Sex-dependent differences in microglial morphology, level of phagocytosis, and expression of pro-inflammatory genes have also been associated with the level of plaque pathology [100], some of which may be ascribed to mitochondrial function [101]. It is therefore reasonable that gene expression changes, including those emanating from microglia, were detectable in female mice but not males in our study, and regardless of baseline or disease-associated changes in DUSP4, to hypothesize that gender-specific interventions may be indicated in the treatment of AD. DUSP4 and the select neuroinflammatory, IFNγ, and ERK pathways regulated in female 5xFAD by its overexpression, that we have identified here, may therefore represent gender-specific, pharmacotherapeutic AD targets.

## Figures and Tables

**Figure 1 cells-11-03880-f001:**
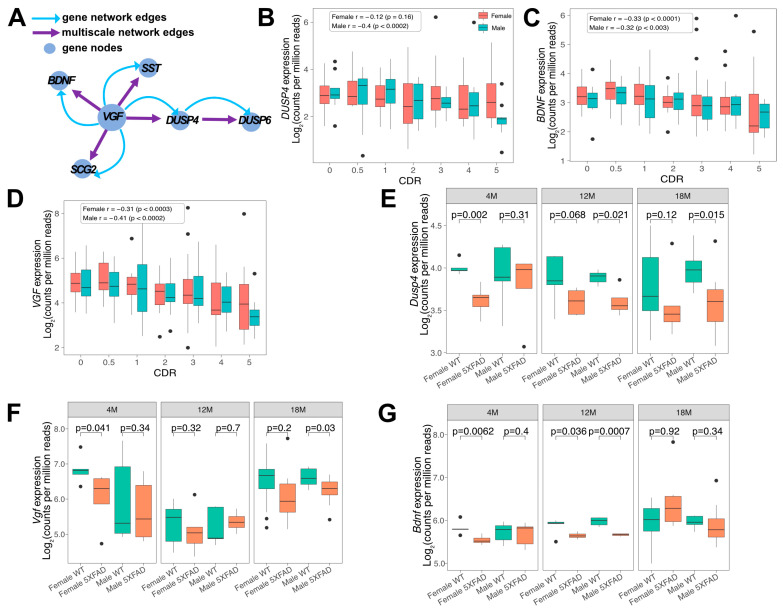
Gene expression changes in female and male hippocampal regions from AD postmortem and 5xFAD brains. (**A**) The representative scheme shows DUSP4 as a network hub or driver located downstream of VGF [14]. The blue nodes are genes. Boxplots (**B**–**D**) show expression of hippocampal (**B**) *DUSP4*, (**C**) *BDNF*, and (**D**) *VGF* mRNAs [Log_2_(counts per million reads)] in AD postmortem brain samples from the Mount Sinai Brain Bank (MSBB) stratified by CDR; *r*, Spearman’s correlation coefficient. Similarly, boxplots E-G compare the expression of hippocampal (**E**) *Dusp4*, (**F**) *Vgf*, and (**G**) *Bdnf* mRNAs among female and male 5xFAD and WT mice at different ages using RNAseq data obtained from the AMP-AD portal (see Appendix A).

**Figure 2 cells-11-03880-f002:**
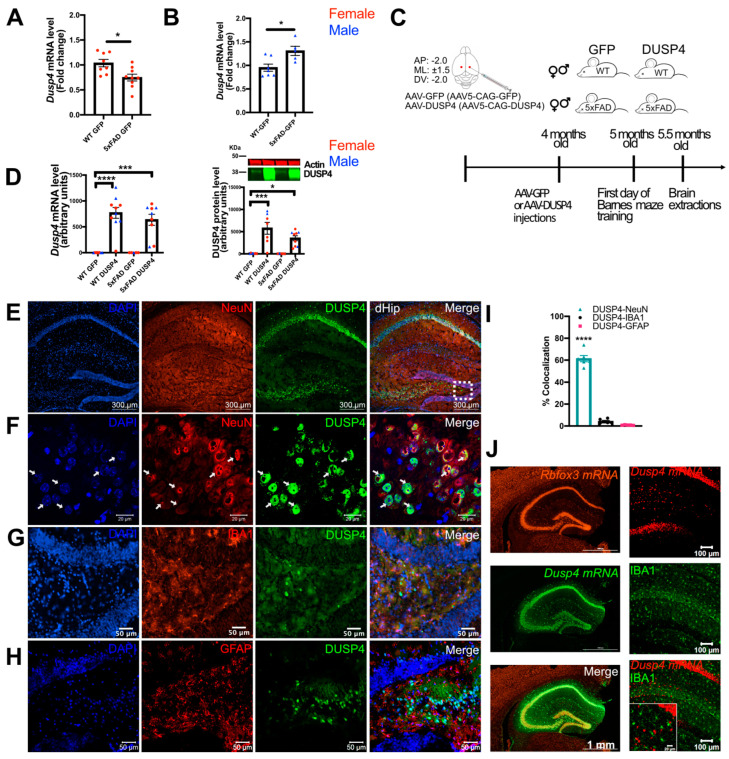
AAV5-mediated overexpression of DUSP4 in dorsal hippocampus (dHc) of 5-month-old 5xFAD and WT mice. (**A**,**B**) *Dusp4* mRNA levels of female (**A**) and male (**B**) 5xFAD were compared to the corresponding sex-matched GFP-overexpressing WT (WT-GFP). (**C**) The coordinates of the injection site, the timeline for the experiment, and the experimental groups are shown. (**D**) RT-PCR (left) and western blot (right) analyses confirmed the overexpression of DUSP4 in 5xFAD and WT, *n* = 5–10 mice/group. (**E**–**H**) Immunohistochemistry for DUSP4 (green), NeuN (red), GFAP (red), IBA1 (red), and DAPI (blue). (**E**) Staining of hippocampi for DUSP4, NeuN, and DAPI. (**F**) The magnified image, covering part of the hilus and DG region indicated by dashed box in Figure 2H, shows co-localization of DUSP4 and NeuN. (**G**) Immunohistochemistry for DUSP4, IBA1, and DAPI. (**H**) Immunohistochemistry for DUSP4, GFAP, and DAPI. (**I**) Graph shows percentage of colocalization using Mander’s correlation coefficient, and the thresholded Mander’s M values corresponding to the fraction of DUSP4 in NeuN (neuron), IBA1 (microglia), or GFAP (astrocyte) analyzed by the JACoP plugin from ImageJ. (**J**) RNAscope images of *Rbfox3*, *Dusp4*, and merge (left panel, from top to bottom), and *Dusp4*, IBA1, and merge (right panel, from top to bottom). Scale bars = 20 µm, 50 µm, 300 µm, or 1 mm as noted on the images. Error bars represent means ± SEM. Statistical analyses were performed using a one-way ANOVA followed by a Tukey’s post-hoc test, * *p* < 0.05, *** *p* < 0.001, **** *p* < 0.0001.

**Figure 3 cells-11-03880-f003:**
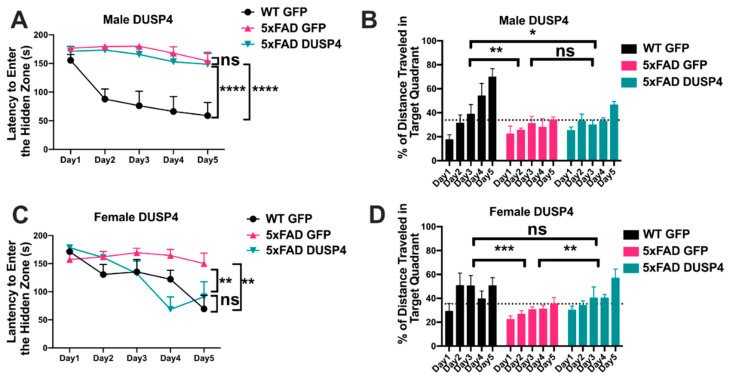
Barnes maze test on 5xFAD mice overexpressing DUSP4. (**A**–**D**) DUSP4-overexpressing or GFP-overexpressing 5xFAD and WT mice were tested in the Barnes maze. Both male (**A**,**B**) and female (**C**,**D**) 5xFAD and WT overexpressing DUSP4 or GFP were trained in the Barnes maze at 5 months of age, *n* = 6–9/group. Training was performed on a 5-day session with two trials per day, and the time (**A**,**C**) spent to enter the hidden tunnel and the percentage of distance traveled in the target quadrant (**B**,**D**) were recorded. Error bars represent means ± SEM. Statistical analyses were performed using a two-way ANOVA followed by a Tukey’s post-hoc test, * *p* < 0.05, ** *p* < 0.01, *** *p* < 0.001, **** *p* < 0.0001; ns nonsignificant.

**Figure 4 cells-11-03880-f004:**
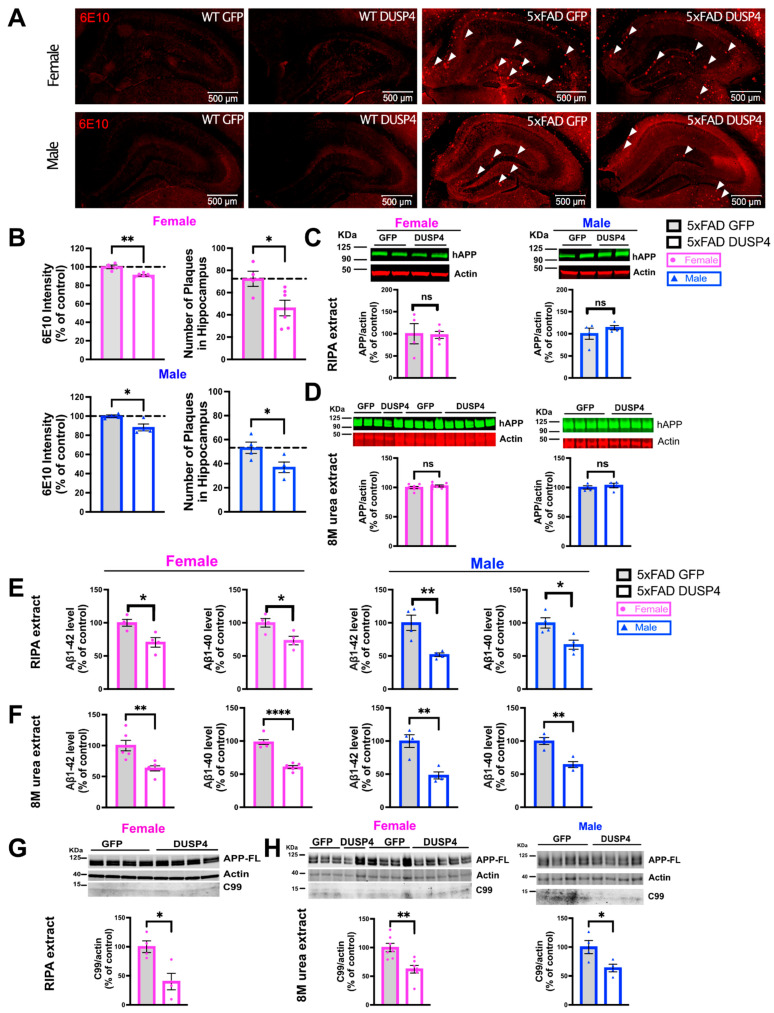
Reduced amyloid plaque load in hippocampi of male and female 5xFAD mice overexpressing DUSP4, compared to 5xFAD-GFP controls. (**A**) Representative images of 6E10 staining for amyloid deposition in hippocampi of male and female 5xFAD mice overexpressing DUSP4 or GFP at 5 months of age. Arrows indicate amyloid plaques in the hippocampi. Scale bar = 500 µm. (**B**) Quantification of intensity and number of 6E10-positive plaques in the hippocampi of female and male 5xFAD and WT mice overexpressing DUSP4 or GFP at 5 months of age, *n* = 4-6 mice per group and per sex with three coronal sections per animal. No 6E10 staining was detected in WT-GFP mice or WT mice overexpressing DUSP4. (**C**) Western blot analysis of human APP-related proteins containing the 6E10 epitope in hippocampi of female and male 5xFAD overexpressing DUSP4 or GFP from RIPA extract, *n* = 4 mice per group. (**D**) Western blot analysis of human APP-related proteins containing the 6E10 epitope in hippocampi of female (left panel) and male (right panel) 5xFAD overexpressing DUSP4 or GFP from 8M urea extract, *n* = 4–7 mice per group. (**E**) ELISA quantification of human Aβ^1–^^40^ and Aβ^1–^^42^ levels in the hippocampi of the mice from the same groups described in (**C**). (**F**) ELISA quantification of human Aβ^1–^^40^ and Aβ^1–^^42^ levels in the hippocampi of the mice from the same groups described in (**D**). (**G**) Western blot analysis of C99 protein in female 5xFAD overexpressing DUSP4 or GFP from RIPA extract. (**H**) Western blot analysis of C99 protein in female and male 5xFAD overexpressing DUSP4 or GFP from 8M urea extract. Error bars represent means ± SEM. Statistical analyses were performed using a Student’s *t*-test, * *p* < 0.05, ** *p* < 0.01, **** *p* < 0.0001; ns = nonsignificant.

**Figure 5 cells-11-03880-f005:**
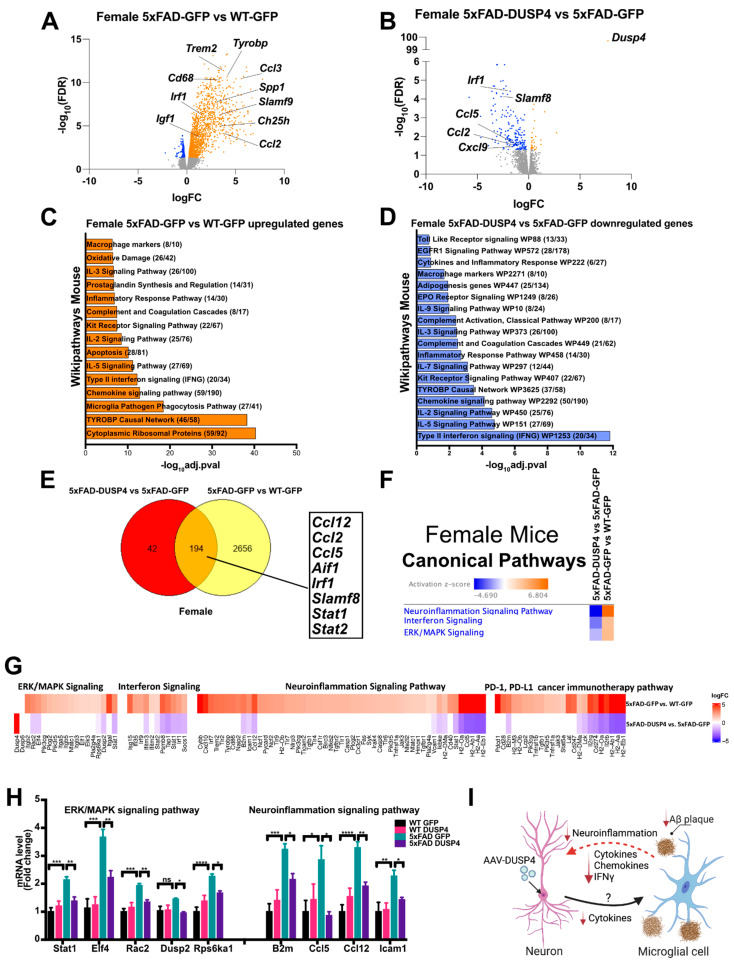
DUSP4 overexpression downregulates DEGs in female 5xFAD, notably in ERK/MAPK, interferon, neuroinflammation and PD-L1/PD-1 pathways. (**A**,**B**) Volcano plot representation of female 5xFAD-GFP vs. WT-GFP, and 5xFAD-DUSP4 vs. 5xFAD-GFP. *n* = 5 mice per group. Threshold for DEGs represented is FDR < 0.05 (orange and blue dots). Enrichment analysis, by wikipathways mouse, of DEGs from (**C**) female 5xFA-GFP vs. WT-GFP, and (**D**) female 5xFAD-DUSP4 vs. 5xFAD-GFP. (**E**) Venn diagram shows the number of DEGs shared in common among 5xFAD-GFP vs. WT-GFP and 5xFAD-DUSP4 vs. 5xFAD-GFP. (**F**) Canonical pathways predicted by IPA based on DEGs of female 5xFAD-DUSP4 vs. 5xFAD-GFP, and 5xFAD-GFP vs. WT-GFP. (**G**) DEGs that are involved in ERK/MAPK, interferon, neuroinflammation and PD-L1/PD-1 pathways predicted by IPA. (**H**) RT-PCR analysis of the genes in the predicted pathways, *n* = 4–6 mice per group. * *p* < 0.05, ** *p* < 0.01, *** *p* < 0.001, **** *p* < 0.0001; ns nonsignificant. (**I**) Crosstalk between neurons and microglia via an unknown mechanism (?) is postulated to be impacted by neuronal DUSP4 overexpression, resulting in suppression of neuroinflammatory and IFNγ signaling (red dashed line) in microglia, leading to reduced expression of cytokines, chemokines and IFNγ. (Created with BioRender.com (accessed on 9 August 2022)).

## Data Availability

All the primary data supporting the conclusions of this study are included in the manuscript and Appendix A. The complete RNA sequence dataset is being deposited at synapse.org (accessed on 9 August 2022).

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
