# Peer review of "Dual-Specificity Protein Phosphatase 4 (DUSP4) Overexpression Improves Learning Behavior Selectively in Female 5xFAD Mice, and Reduces β-Amyloid Load in Males and Females"

_cells, 2022, doi:10.3390/cells11233880_

Round 1

Reviewer 1 Report

In this study, the authors investigated hippocampal expression of DUSP4 in male AD subjects and the effect of bilateral hippocampal injection of AAV5 -DUSP4 gene in 5xFAD mice on learning behavior and neuropathology.  They found that DUSP4 expression is decreased in AD patients and correlates with increased CDR scores as well as in 5xFAD mice. Delivery of DUSP4 gene to the hippocampus of 5 mth old 5xFAD mice reduced amyloid load in males and females , but  Barnes Maze performance was rescued only in females. RNA-seq analysis revealed that there were DEGSs  primarily related to neuroinflammation , and ERK/MAPK pathways between AAV5-DUSP4 treated animals and untreated 5xFAD mice. However genes encoding neuroprotective proteins such as bdnf and vgf were surprisingly downregulated.  This study has raised more questions than answers as to the role of DUSP4 in mitigating AD neuropathology  and cognitive dysfunction. The effect of AAV-DUSP4  gene delivery is not very strong in rescuing cognitive function and is gender specific.  Therefore DUSP4 is not a good target for AD therapy. AD is a very complex disease and perhaps the RNA seq data suggest that DUSP4 may play a significant role in modulating neuroinflammatory pathways. However, as the authors pointed out, since DUSP4 is expressed in neurons, its effects on neuroinflammation has to be transmitted to microglia and the mechanism is unclear.  

Specific comments:

1.The study would be stronger if more than 1 behavioral test was done.  Sometimes while not significant for Barnes maze test for the males, another test such as Novel Object Recognition Test may show significance for the males as well.  I recommend adding this test to the study. 

2. Effect on hyperphosphorylation of tau should also be examined after AAV-DUSP4 injection in the 5xFAD mice. The results  will determine if DUSP4  has any effect on tau phosphorylation and might further shed light on the difference in the alleviation of cognitive dysfunction by DUSP4 in males versus females.

3. If DUSP4 overexpression really has an effect on decreasing neuroinflammation as suggested from the RNA seq analysis, then the authors should provide some supportive evidence by quantification of a microglia marker and count the number of activated microglia cells in the hippocampus of  AAV-DUSP4 treated vs. AAV-GFP in AD mice. One would expect to see a decrease in these 2 parameters in the DUSP4 treated animals.  

4. The A-beta 1-42 was decreased with DUSP4 overexpression but does not correlate well with rescue of behavioral deficits in males. It be important to determine the proportion of A-beta 1-42 that is soluble vs insoluble to evaluate  if there is a difference between males and females. That may have given a better correlation with behavior as males may have more of the insoluble form versus females, even though total amyloid load is down for both with AAV-DUSP4 treatment.

5. I suggest shortening the discussion which is heavily focused on the  DEGs from RNA-seq  to build hypotheses. These gene changes represent a snap shot in time and may not reflect the protein levels which is more reliable. 

Reviewer 2 Report

This study by Pan et al., investigated the role of DUSP4 in the pathogenesis of Alzheimer’s disease (AD) by using 5XFAD, an amyloid mouse model. DUSP4 is one of down-stream genes in the VGF network that they previously reported. By using previous dataset, they confirmed that DUPS4 was generally down-regulated in 5XFAD mice as well as AD brains. They bilaterally injected AAV5-DSUP4 into the dorsal hippocampus of WT and 5XFAD mice at 4 months of age, resulting in expression in neurons at 6 moths of age. At 5 months old, they observed improvement of learning score of Barnes maze test in female 5XFAD mice, but not in male 5XFAD mice. They observed reduction of Abeta deposition and levels in both male and female 5XFAD mice with DUSP4 expression. They found DUSP4 expression down-regulates most VGF-associated genes by doing real-time PCR. Also, they conducted bulk RNA sequence in female groups, and observed that DUSP4 reduced genes related to neuroinflammation, programmed cell death, and EPK/MAPK signaling pathways. While these results are interesting, the mechanism is speculative. Figures are too many, some of which are not so important to draw the conclusion. Overall, the current manuscript is immature and should be further revised.

1.    The title maybe misleading. As the improvements of learning behavior is female-specific, authors should add “female”, “in a sex-dependent manner” or some related words.

2.    Authors should check APP-CTF (C99) to confirm that DUSP4 really does not affect Abeta production in 5XFAD mice.

3.    Abeta levels should be measured by formic acid, guanidine or other chaotropic extraction, not RIPA.

4.    The mechanism is too speculative. In Fig. 6H, there was no change in these neuroinflammatory or EPK/MAPMK genes in WT-DUSP4, compared to WT-GFP. If DUSP4 really has some biological effects that connect between neuron and microglia as they proposed in Fig. 6I, the effects should also be evident in WT-DUSP4. Rather, this data suggests that DUSP4 affects these genes through reducing Abeta deposition in 5XFAD mice.

5.    Authors should address more about the direct effects of DUSP4, which is one of dual-specify phosphatase family, and is known to have several substrates, such as ERK, JNK, p38. Examination of these phosphorylation would provide insights into the direct effects of DUSP4. 

6.    Figures are too many and not easy to follow-up. It is better to move some of them to supplementary data.

Round 2

Reviewer 1 Report

The authors have addressed all the comments of the reviewer satisfactory . The additional data provided has improved the manuscript. 

Reviewer 2 Report

Authors have well revised the manuscript according to my previous comments. One minor point is better to do additional discussion about reduced C99 levels in terms of potential changes of Abeta production.
